# Enhancing Post-Treatment Visual Acuity Prediction with Multimodal Deep Learning on Small-scale Clinical and OCT Datasets

**Matthew Anderson**[1]                                    M.ANDERSON10@NEWCASTLE.AC.UK
**Veronica Corona**[2]                                         VERONICA.CORONA@ROCHE.COM
**Agnieszka Stankiewicz**[1,3]                   AGNIESZKA.STANKIEWICZ@NEWCASTLE.AC.UK
**Maged Habib**[4,5]                                            MAGED.HABIB2@NHS.NET
**David H. Steel**[4,5]                                     DAVID.STEEL@NEWCASTLE.AC.UK
**Boguslaw Obara**[1,4]                                 BOGUSLAW.OBARA@NEWCASTLE.AC.UK

[1] *School of Computing, Newcastle University, United Kingdom.*

[2] *Roche Products Limited, Welwyn Garden City, United Kingdom.*

[3] *Institute of Automatic Control and Robotics, Poznan University of Technology, Poland.*

[4] *Biosciences Institute, Newcastle University, United Kingdom.*

[5] *Sunderland Eye Infirmary, National Health Service, Sunderland, United Kingdom.*

**Editors:** Accepted for publication at MIDL 2025

## Abstract

Predicting visual acuity (VA) outcomes after treatment in diabetic macular edema (DME) is crucial for optimizing patient management but remains challenging due to the heterogeneity of patient responses and the limited availability of comprehensive datasets. While existing predictive models have shown promise, their clinical deployment is hindered by their reliance on large training datasets that are often unavailable in real-world settings. We address this challenge by developing a multimodal deep learning framework specifically designed for small-scale clinical cohorts. Our approach integrates optical coherence tomography (OCT) images with carefully selected clinical parameters through a cross-modal fusion architecture that leverages attention mechanisms to enhance feature interaction and predictive accuracy. We validate our framework across two clinically distinct real-world cohorts: treatment-naïve patients ($n = 35$) receiving intensive anti-VEGF therapy and chronically treated patients ($n = 20$) receiving sustained-release corticosteroid implants. This approach achieves mean absolute errors in post-treatment VA prediction of $3.07 \pm 0.82$ and $4.20 \pm 2.79$ Early Treatment Diabetic Retinopathy Study (ETDRS) letters, respectively, falling within the acceptable range of clinical measurement variability and meeting thresholds for statistically significant visual change detection with $\geq 90\%$ confidence. This work demonstrates that appropriately designed multimodal architectures can achieve clinically meaningful prediction accuracy even with limited datasets, offering a practical foundation for personalized DME management in typical clinical settings where large datasets are unavailable.

**Keywords:** Deep Learning, Multimodal Learning, Visual Acuity, Optical Coherence Tomography (OCT), Diabetic Macular Edema

## 1. Introduction

DME is a prevalent complication of diabetic retinopathy and a leading cause of vision impairment among individuals with diabetes worldwide (Peto and Tadros, 2012). The condition arises from hyperglycemia-induced damage to retinal blood vessels, leading to fluid accumulation in the macula, which can result in vision loss or blindness (Davidson et al., 2007). While anti-VEGF therapies are effective in improving visual outcomes (Stefanini et al., 2014), patient responses remain highly variable (Chen et al., 2019), necessitating predictive tools to guide personalized treatment plans. Changes in VA, measured using ETDRS letter scores, are a common benchmark for evaluating treatment success.

Multimodal deep learning has shown promise for predicting treatment outcomes by integrating diverse data types. For DME, treatment outcomes are influenced not only by imaging characteristics but also by systemic factors and treatment history (Bressler et al., 2012; Dugel et al., 2019). Lin et al. (Lin et al., 2024) demonstrated improved outcome prediction for glaucoma by combining operative notes with health records, while Wen et al. (Wen et al., 2023) achieved robust VA prediction ($R^2 = 0.80$) using OCT and clinical data fusion. For anti-VEGF therapy in DME, Liu et al. (Liu et al., 2021) demonstrated that an ensemble machine learning system combining deep learning and classical ML models could accurately predict post-treatment outcomes (central foveal thickness and BCVA) in patients receiving anti-VEGF injections. However, these methods often rely on large, well-curated datasets, limiting their generalizability to real-world clinical settings.

Developing reliable predictive models for DME presents significant challenges, including limited availability of large, high-quality datasets (Anderson et al., 2023; Whang et al., 2023), the complexity of integrating heterogeneous clinical and imaging data (Mårtensson et al., 2020), and compliance with privacy regulations (Williamson and Prybutok, 2024). Despite significant advancements, translating deep learning methods into clinical practice for small, diverse patient populations remains a major hurdle.

This study addresses the challenge of predicting post-treatment VA in small cohorts of patients with DME undergoing treatment using a novel multimodal deep learning framework. The proposed approach integrates OCT imaging and clinical data to improve prediction accuracy, even with limited datasets. Our main contributions are as follows:

1. Careful clinical feature selection (Sec. 3.2), using statistical methods to identify robust predictors, ensuring the model focuses on clinically relevant factors.

2. A hybrid neural network architecture (Sec. 3.3) combining an EfficientNet-B0-based image encoder with a feedforward network for clinical data. The framework integrates these modalities through a fusion network for effective multimodal prediction.

3. Demonstration of the superiority of the multimodal approach over single-modality methods (Sec. 4), leveraging complementary data sources to address the challenges of small datasets.

## 2. Materials

This study utilized data from two prospective clinical trials at Sunderland Eye Infirmary, UK: the DIME Study (NIHR CPMS ID 48908) of treatment-naïve patients receiving intensive

anti-VEGF therapy, and the INDEX Study (IRAS ID 281058) of chronic DME patients receiving fluocinolone acetonide intravitreal implants. Key demographics are presented in Table 1.

Table 1: Demographic and clinical characteristics of study participants.

| Characteristic | Study Cohort | |
| --- | --- | --- |
| | DIME | INDEX |
| Total eyes | 35 | 20 |
| Age (years) | $56.83 \pm 13.70$ | $68.75 \pm 5.91$ |
| Gender (M/F) | 26/9 | 15/5 |
| DM duration (years) | $18.23 \pm 8.50$ | $20.60 \pm 9.70$ |
| DM type (1/2) | 13/22 | 2/18 |
| VA (ETDRS letters) | | |
| Baseline | $69.50 \pm 9.28$ | $64.90 \pm 10.26$ |
| Final follow-up* | $75.60 \pm 9.49$ | $67.95 \pm 9.70$ |

*Final follow-up was at 12 months for DIME study and 4-6 months for INDEX study.
DM = Diabetes Mellitus.

The DIME Study included 35 treatment-naïve diabetic patients with active DME (central macular thickness $> 400$ µm). Exclusion criteria included media opacities, laser scars within 1000 µm of foveal centre, concurrent macular disease, and vitreo-macular interface abnormalities. OCT imaging (Heidelberg Spectralis) parameters: scan angle $30° \times 15°$, 145 lines, 30 µm spacing, $1536 \times 496$ pixels. Two eyes were excluded due to missing OCT volumes.

The INDEX study enrolled 20 chronic DME patients with $\geq 12$ months' disease history and suboptimal dexamethasone response. Exclusion criteria included prior dexamethasone response ($\geq 20\%$ CST reduction at 4+ months), other macular edema causes, macular pathology precluding improvement, pre-existing glaucoma/IOP issues ($> 25$ mmHg requiring multiple treatments), and active ocular infection. OCT imaging (Heidelberg Spectralis) parameters: scan angle $20° \times 20°$, 49 lines, 120 µm spacing, $512 \times 496$ pixels.

## 3. Methods

### 3.1. Image Preprocessing

OCT volumes from both datasets underwent standardization for resolution and dimensions. For DIME, horizontal pixel resolutions (range: 5.43-11.92 µm) were standardized via Winsorization (Dixon and Yuen, 1974), with outliers beyond $\pm 1.5$ µm from the mean (6.089 µm) scaled accordingly, while axial resolution was constant (3.87 µm). OCT volumes (145 B-scans, width $\times$ height: $1536 \times 496$ pixels) were resized to $512 \times 512$ pixels. For INDEX, horizontal pixel resolutions (10.61-11.88 µm) and axial resolution (3.87 µm) required no rescaling, and B-scans (49 slices, width $\times$ height: $512 \times 496$ pixels) were center-cropped to $496 \times 496$ pixels.

Image augmentations included rotations ($\pm 8$-11°), horizontal flips, shifts, Gaussian noise (variance 0.1-8.6), and coarse dropout, with parameters optimized per dataset.

### 3.2. Clinical Feature Selection

Feature selection combined statistical and machine learning approaches: Pearson correlation (Pearson, 1895) to assess linear relationships, random forest importance (Breiman,

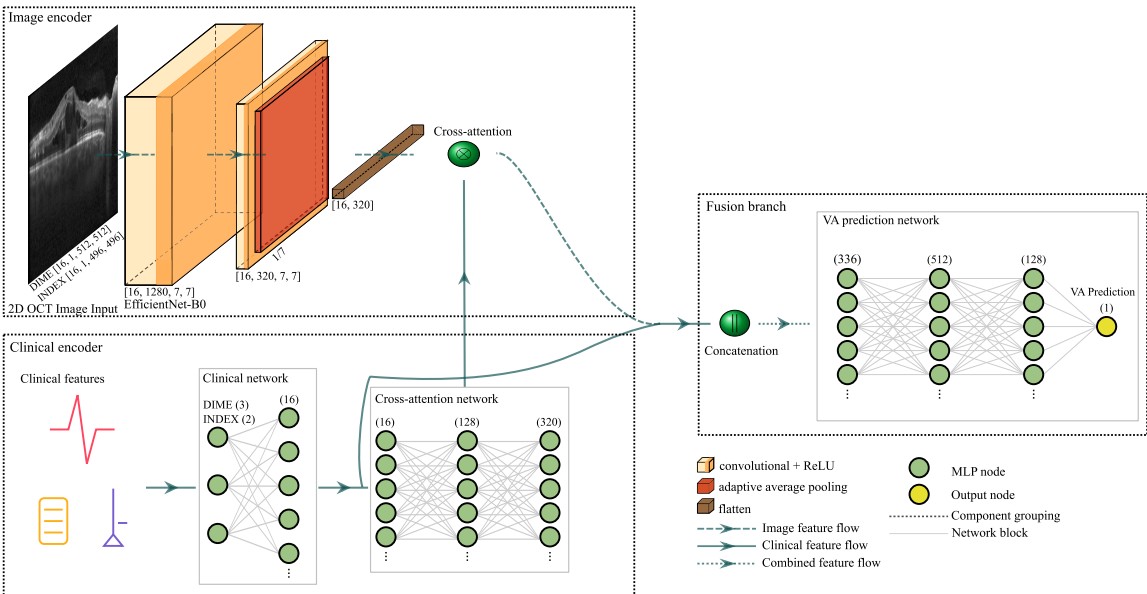

Figure 1: Architecture of the proposed multimodal network combining OCT imaging and clinical features, with dimensions shown at each major processing stage.

2001) for non-linear interactions, F-test ANOVA (Fisher, 1925) for group differences, and mutual information analysis (Shannon, 1948) for general statistical dependencies. Following established guidelines for small datasets (Peduzzi et al., 1996), we limited selection to one feature per 10 observations.

For the DIME dataset, this selection process identified three clinical features: baseline VA (ETDRS), patient age (years), and DM duration (years), which consistently ranked as the most predictive parameters across our analysis methods. For the INDEX dataset, two clinical features were selected: baseline VA (ETDRS) and DM duration (years). These features were chosen based on their consistent performance across feature importance methods and their established clinical relevance in DME treatment response. Comprehensive feature evaluation details are provided in Appendix Tables 5 and 6.

### 3.3. Model Architecture

Let $\mathbf{X}_I \in \mathbb{R}^{b \times 1 \times h \times w}$ denote the input 2D OCT image (B-scan), where $h$ and $w$ are the height and width of the input image in pixels, and the single channel corresponds to grayscale images. $\mathbf{X}_C \in \mathbb{R}^{b \times p}$ represents the clinical features, where $b = 16$ is the batch size, and $p$ is the number of clinical parameters used ($p = 3$ for DIME: baseline VA (ETDRS), age (years), DM duration (years), and $p = 2$ for INDEX: baseline VA (ETDRS), DM duration (years)). The proposed architecture integrates these inputs through three primary components: an image encoder, a clinical encoder, and a prediction network, as shown in Figure 1.

The image encoder employs an EfficientNet-B0 backbone pre-trained on ImageNet (Deng et al., 2009), modified to process grayscale input by averaging the pre-trained RGB channel weights in the first convolutional layer. During training, only blocks 6 and 7 were fine-tuned, while earlier layers were frozen to reduce computational cost. The final pooling

layer flattens the spatial dimensions to produce a feature vector of 1280 channels, followed by dimensionality reduction using a 1x1 convolutional layer with batch normalization and ReLU activation, projecting the image features to $\mathbf{h}_I \in \mathbb{R}^{b \times 320}$.

The clinical features $\mathbf{X}_C$ are processed through fully connected layers with LayerNorm and GELU activation, producing transformed clinical features $\mathbf{h}_C \in \mathbb{R}^{b \times 16}$.

Cross-modal attention integrates information from both modalities by modulating the image features based on the clinical features:

$$\mathbf{h}_{att} = \mathbf{h}_I \odot \sigma(\mathbf{W}_3 \cdot \text{GELU}(\mathbf{W}_2 \cdot \mathbf{h}_C)), \tag{1}$$

where $\sigma$ is the sigmoid activation function, $\odot$ denotes element-wise multiplication, and $\mathbf{W}_2 \in \mathbb{R}^{128 \times 16}$ and $\mathbf{W}_3 \in \mathbb{R}^{320 \times 128}$ are trainable weight matrices.

For each B-scan $s$, the attended image features $\mathbf{h}_{att}$ are concatenated with the clinical features $\mathbf{h}_C$:

$$\hat{\text{VA}}_s = f_P([\mathbf{h}_{att}; \mathbf{h}_C]) \in \mathbb{R}^{b \times 1}, \tag{2}$$

where $f_P(\cdot)$ denotes the prediction network, a three-layer fully connected module with input-output dimensions of $336 \to 512 \to 128 \to 1$, using LayerNorm, GELU activation, and dropout (rate = 0.3).

During training and inference, the model maintains patient-level consistency by replicating clinical features across all B-scans belonging to the same patient's OCT volume. Let $\mathcal{S} = \{1, \ldots, M\}$ be the set of B-scan indices in a patient's volume. The final VA prediction for each patient is obtained by averaging these B-scan predictions:

$$\hat{\text{VA}}_{\text{patient}} = \frac{1}{M} \sum_{s \in \mathcal{S}} \hat{\text{VA}}_s. \tag{3}$$

This architecture leverages cross-modal attention and fusion to integrate OCT imaging biomarkers with clinical metadata, enhancing interpretability and generalization for robust post-treatment VA prediction on small, heterogeneous datasets.

### 3.4. Training and Evaluation

The model was trained using AdamW optimization (Loshchilov and Hutter, 2019) with cosine annealing warm restarts (Loshchilov and Hutter, 2017) (*learning rate* $= 10^{-3}$, *weight decay* $= 0.01$). Training was performed using Huber loss (Huber, 1992), with $\delta = 1.0$ (the threshold at which the loss transitions from quadratic to linear), and early stopping was applied after 10 epochs without improvement.

Evaluation used stratified 5-fold cross-validation, with stratification based on post-treatment VA outcomes (low: $<71$, medium: 71-82, high: $>82$ ETDRS letters). These thresholds were chosen with reference to $+0.30$ logMAR (70 ETDRS letters), a key minimum visual standard required to hold a UK driver's license (Rae et al., 2016). Performance was assessed using standard regression metrics: mean absolute error (MAE), root mean square error (RMSE), and coefficient of determination ($R^2$), where predictions for each patient were computed by averaging batch predictions according to Equation 3.

## 4. Results

Table 2: Comparative analysis of VA prediction models.

| Model | DIME Dataset | | | INDEX Dataset | | |
|---|---|---|---|---|---|---|
| | MAE↓ | RMSE↓ | $R^2$↑ | MAE↓ | RMSE↓ | $R^2$↑ |
| *Clinical Data Models* | | | | | | |
| Linear Regression | $4.73 \pm 0.98$ | $6.05 \pm 1.37$ | $0.55 \pm 0.17$ | $6.89 \pm 3.99$ | $7.67 \pm 3.90$ | $0.25 \pm 0.49$ |
| Random Forest | $5.17 \pm 1.41$ | $7.58 \pm 2.29$ | $0.25 \pm 0.51$ | $6.09 \pm 3.51$ | $6.84 \pm 4.09$ | $0.33 \pm 0.52$ |
| Neural Network | $7.78 \pm 1.51$ | $9.82 \pm 2.69$ | $-0.21 \pm 0.41$ | $8.18 \pm 1.77$ | $10.35 \pm 4.25$ | $-0.31 \pm 0.48$ |
| *OCT Image Models* | | | | | | |
| EfficientNet-b0 | $5.29 \pm 1.53$ | $6.26 \pm 1.74$ | $0.53 \pm 0.17$ | $6.87 \pm 1.98$ | $8.34 \pm 2.26$ | $0.14 \pm 0.22$ |
| ResNet-50 | $6.22 \pm 2.29$ | $7.35 \pm 2.52$ | $0.33 \pm 0.29$ | $6.27 \pm 1.72$ | $7.80 \pm 1.33$ | $0.21 \pm 0.25$ |
| *Multimodal Models* | | | | | | |
| **Proposed Model** | $\mathbf{3.07 \pm 0.82}$ | $\mathbf{4.03 \pm 1.12}$ | $\mathbf{0.77 \pm 0.16}$ | $\mathbf{4.20 \pm 2.79}$ | $\mathbf{4.87 \pm 3.52}$ | $\mathbf{0.61 \pm 0.36}$ |
| *Reference Models (External Datasets)* | | | | | | |
| Ensemble ML (Liu et al., 2021)* | 6.5 | 10.0 | 0.68 | — | — | — |
| OCT-based DL (Wen et al., 2023)† | 3.5 | 5.5 | 0.80 | — | — | — |

Note: Values presented as mean ± standard deviation across 5-fold cross-validation. ↑ indicates higher is better, ↓ indicates lower is better.
All error metrics are in ETDRS letters. *Tested on GDPH/ZHSMU dataset. †Tested on iERM dataset.

Our proposed multimodal approach outperformed single-modality baselines using either clinical features alone (linear regression, random forest, neural network with the same architecture as the clinical network model but with a regression head) or OCT images alone (ResNet-50 and EfficientNet-B0). It achieved superior performance across all metrics (Table 2), with an MAE of $3.07 \pm 0.82$ ETDRS letters ($R^2$: $0.77 \pm 0.16$) on DIME and $4.20 \pm 2.79$ ($R^2$: $0.61 \pm 0.36$) on INDEX, significantly outperforming both clinical-only (best MAE: 4.73) and imaging-only models (best MAE: 5.29) in both treatment-naïve and chronic DME cases. The proposed model also demonstrated better error metrics than reference models from Liu et al. (Liu et al., 2021) and comparable performance to Wen et al. (Wen et al., 2023), though these were evaluated on different datasets, so direct comparison should be made cautiously. Detailed per-fold performance metrics are provided in Appendix Table 4.

Interpretability analysis through Grad-CAM revealed distinct spatial attention distributions (Figure 2). Treatment-naïve cases (DIME) demonstrated focal activation patterns localizing to regions of intraretinal fluid (IRF), while chronic cases (INDEX) exhibited broader attention distribution across areas of edema and structural alteration, consistent with established pathological progression patterns (Sakini et al., 2024).

Systematic error patterns varied across VA ranges (Figure 3a,b). In DIME (Figure 3(a)), low VA cases ($< 71$ ETDRS letters, $n = 10$) showed over-prediction ($2.24 \pm 3.61$ ETDRS letters), while high VA cases ($> 82$ ETDRS letters, $n = 11$) showed under-prediction ($-2.06 \pm 3.07$ ETDRS letters). In INDEX (Figure 3(b)), low VA predictions ($n = 12$) were balanced ($1.66 \pm 5.56$ ETDRS letters), while higher VA ranges showed under-prediction bias ($n = 6$, $-1.39 \pm 1.34$ ETDRS letters; $n = 2$, $-11.86 \pm 2.74$ ETDRS letters).

Feature importance analysis through Integrated Gradients quantified the relative contributions of imaging and clinical features on the DIME dataset (Figure 4). OCT features provided the strongest predictive signal (mean $= 8.07 \pm 4.21$), complemented by baseline VA as the primary clinical indicator (mean $= 5.78 \pm 1.95$).

Table 3: Ablation study of model components.

| Model Configuration | DIME Dataset | | | INDEX Dataset | | |
|---|---|---|---|---|---|---|
| | MAE↓ | RMSE↓ | R²↑ | MAE↓ | RMSE↓ | R²↑ |
| **Proposed Model** | 3.07 ± 0.82 | 4.03 ± 1.12 | 0.77 ± 0.16 | 4.20 ± 2.79 | 4.87 ± 3.52 | 0.61 ± 0.36 |
| *Clinical Features* | | | | | | |
| All Clinical Features Variation (11/18) | 4.85 ± 1.67 | 6.71 ± 2.26 | 0.42 ± 0.31 | 6.56 ± 2.97 | 8.57 ± 4.85 | 0.09 ± 0.59 |
| Replace Baseline VA with: | | | | | | |
|   - IRF Cysts | 5.45 ± 1.18 | 7.14 ± 1.83 | 0.35 ± 0.20 | — | — | — |
|   - Baseline IOP (mmHg) | — | — | — | 7.54 ± 1.95 | 9.02 ± 4.32 | 0.05 ± 0.15 |
| *Model Architecture Variation* | | | | | | |
| Without Attention Mechanism | 3.84 ± 1.13 | 5.06 ± 1.46 | 0.66 ± 0.19 | 4.57 ± 2.31 | 6.13 ± 4.43 | 0.59 ± 0.30 |

Note: DIME contains 11 clinical features while INDEX contains 18 clinical features for the all clinical features configuration. IOP = Intraocular Pressure.

To further assess the contribution of individual components in our multimodal framework, we conducted an ablation study (Table 3). When using all available clinical features

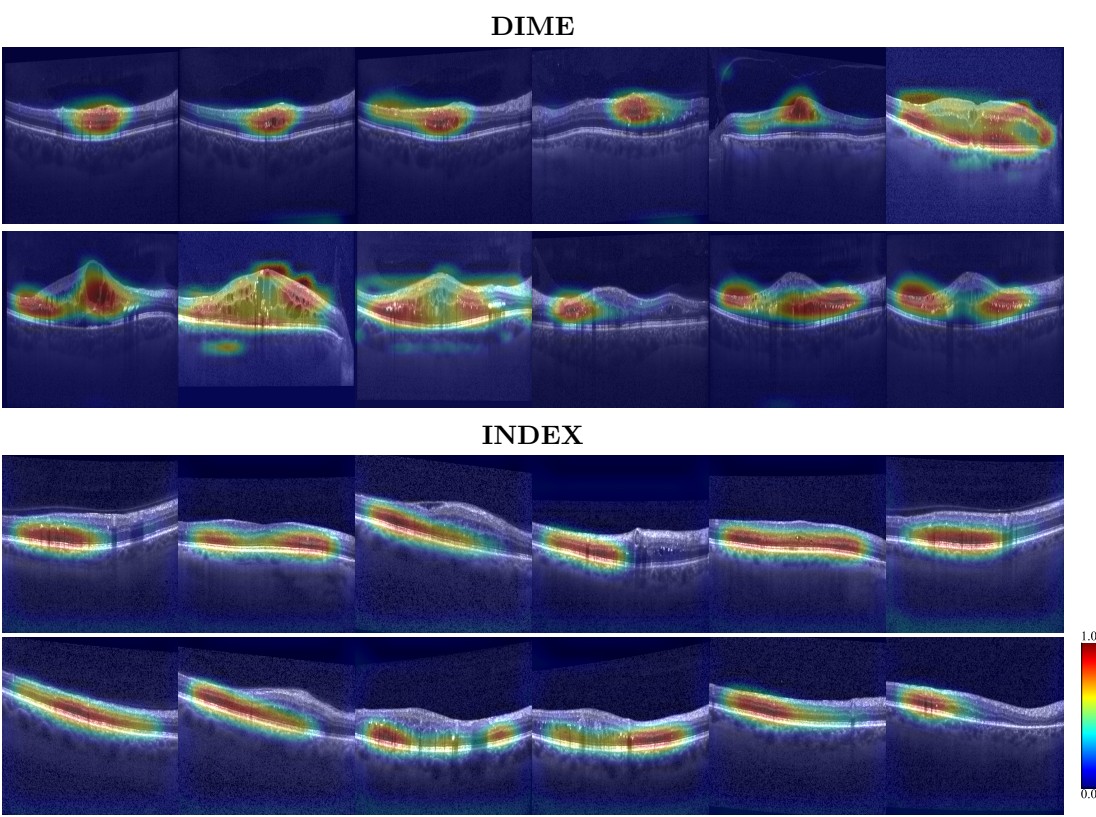

Figure 2: Guided Grad-CAM activation maps for randomly chosen OCT images in the DIME (top) and INDEX (bottom) datasets, highlighting regions of interest identified by the model. The color bar indicates activation magnitude from 0.0 (blue) to 1.0 (red). Images were selected randomly to illustrate typical model interpretations across the datasets.

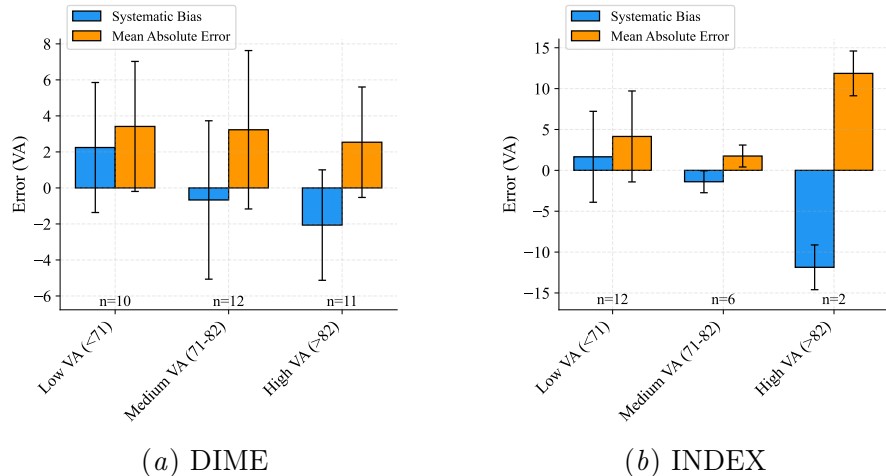

(a) DIME          (b) INDEX

Figure 3: Error analysis by post-treatment VA range. Blue bars show systematic bias (direction of error); orange bars show MAE. Patient counts (n) are displayed below each group. Positive values indicate VA overestimation; negative values show underestimation.

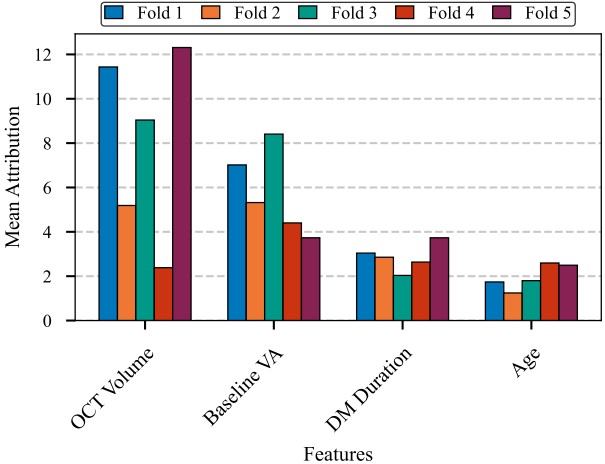

Figure 4: Mean feature attribution magnitudes across five-fold validation showing relative importance of imaging and clinical predictors on DIME dataset. OCT volume measurements show the highest variability, suggesting dataset-specific learning patterns.

(11 for DIME, 18 for INDEX) rather than our selected subset, model performance degraded significantly (MAE increased by 1.78 and 2.36 ETDRS letters for DIME and INDEX, respectively), demonstrating the effectiveness of our feature selection approach in mitigating overfitting on small datasets.

Replacing baseline VA with alternative features (IRF cysts for DIME; baseline IOP for INDEX) resulted in substantial performance deterioration, confirming baseline VA as a critical predictor. Additionally, removing the cross-modal attention mechanism increased prediction error by 25% for DIME and 9% for INDEX, highlighting the importance of

modality-specific feature weighting in our architecture. These findings validate our design choices for both feature selection and architectural components.

## 5. Discussion

Our multimodal framework achieves clinically meaningful error margins, maintaining average prediction errors below 5 ETDRS letters across both datasets - within the threshold for statistically detectable change ($\geq 90\%$ confidence) in eyes with VA better than 20/100 (Beck et al., 2007). The enhanced performance in DIME (MAE: $3.07 \pm 0.82$) compared to INDEX (MAE: $4.20 \pm 2.79$) can be attributed to higher OCT resolution and larger sample size. Analysis across VA ranges revealed distinct prediction patterns: DIME exhibited regression toward the mean (over-prediction in low VA, under-prediction in high VA ranges), while INDEX demonstrated under-prediction bias in higher VA ranges, likely influenced by its smaller sample size ($n=2$ for high VA). Our ablation studies confirm the critical importance of baseline VA as a clinical predictor, with its replacement significantly degrading performance in both datasets. Additionally, the cross-modal attention mechanism proved essential for accurate predictions, particularly for treatment-naïve patients (25% error increase when removed). These findings validate our approach to feature selection and architectural design for small clinical datasets. Feature attribution analysis validates OCT volumes as the primary predictive signal, complemented by clinical parameters, aligning with findings on the relevance of both anatomical and clinical factors in DME progression (Antonetti et al., 2006). Grad-CAM visualization demonstrated anatomically relevant attention patterns corresponding to established OCT biomarkers (Zur et al., 2018; Sun et al., 2014), with differential activation between treatment-naïve and chronic cases. Limitations include systematic biases requiring range-specific calibration, small dataset size necessitating k-fold cross-validation, and dataset heterogeneity in OCT protocols and treatment modalities complicating cross-cohort comparison. Prospective validation with expert-graded OCT characteristics would help establish clinical utility.

## 6. Conclusion

Our multimodal deep learning approach successfully predicted post-treatment VA in DME patients with mean absolute errors of $3.07 \pm 0.82$ ETDRS letters (treatment-naïve) and $4.20 \pm 2.79$ ETDRS letters (chronically treated). These errors, below the clinically significant threshold of 5 ETDRS letters, demonstrate potential for informing treatment decisions. Ablation experiments validated our feature selection approach and cross-modal attention mechanisms, while Grad-CAM analysis revealed pathology-specific attention patterns between patient cohorts. While these findings highlight the promise of multimodal deep learning for small datasets, further validation with multi-centre studies and external datasets is needed. Future research should optimize architectures for small datasets, improve generalizability, and address systematic biases, enabling valuable clinical insights and personalized treatment planning for DME patients.

## Code and Data Availability

The code is publicly available at [https://github.com/muanderson/VA_MM_DL](https://github.com/muanderson/VA_MM_DL). The data is available upon request.

## Acknowledgments

This work was supported by the Engineering and Physical Sciences Research Council (EP-SRC) and industrial partners of the Centre for Doctoral Training in Cloud Computing for Big Data (EP/L015358/1). It was further facilitated through a collaborative working agreement between Roche Products Limited, The University of Newcastle Upon Tyne, and South Tyneside and Sunderland NHS Foundation Trust. M-GB-00021023 | December 2024.

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

## Appendix A. Additional Figures and Tables

Table 4: Per-fold performance of the proposed model.

| Fold | DIME Dataset | | | INDEX Dataset | | |
|---|---|---|---|---|---|---|
| | MAE↓ | RMSE↓ | $R^2$↑ | MAE↓ | RMSE↓ | $R^2$↑ |
| Fold 1 | 3.72 | 4.67 | 0.81 | 2.89 | 3.14 | 0.84 |
| Fold 2 | 3.12 | 4.11 | 0.78 | 0.77 | 0.86 | 0.99 |
| Fold 3 | 1.93 | 2.43 | 0.95 | 5.12 | 5.86 | 0.25 |
| Fold 4 | 2.41 | 3.27 | 0.85 | 9.06 | 11.17 | 0.13 |
| Fold 5 | 4.17 | 5.66 | 0.47 | 3.17 | 3.35 | 0.87 |
| Mean | 3.07 | 4.03 | 0.77 | 4.20 | 4.87 | 0.61 |
| Std. Dev. | 0.82 | 1.12 | 0.16 | 2.79 | 3.52 | 0.36 |

Table 5: Predictive importance of clinical features for post-treatment VA in DIME using different feature selection methods. Features selected for the proposed approach are indicated in bold.

| Feature | Feature Selection Metrics | | | |
|---|---|---|---|---|
| | Pearson Correlation | F-test ANOVA | Mutual Information | Random Forest Importance |
| *Primary Clinical Parameters* | | | | |
| **Baseline VA (ETDRS)** | 0.5220 | 11.6093 | 0.2965 | 0.2625 |
| **Age (years)** | 0.3575 | 4.5426 | 0.0947 | 0.1370 |
| **DM duration (years)** | 0.2652 | 2.3445 | 0.3366 | 0.2519 |
| *Anatomical Features* | | | | |
| IRF Cysts | 0.3374 | 3.9822 | 0.1149 | 0.0322 |
| SRF Fluid | 0.2614 | 2.2739 | 0.0251 | 0.0143 |
| Deep Haemorrhages Within Macula | 0.1527 | 0.7396 | 0.0000 | 0.0200 |
| Exudates Within Macula | 0.1309 | 0.5404 | 0.0164 | 0.0115 |
| *Morphometric Measurements* | | | | |
| Baseline Macular Volume ($mm^3$) | 0.0705 | 0.1547 | 0.0565 | 0.1737 |
| Baseline CST (µm) | 0.0624 | 0.1210 | 0.0880 | 0.0875 |
| *Demographic Factors* | | | | |
| DM type | 0.1819 | 1.0611 | 0.0292 | 0.0023 |
| Gender | 0.1604 | 0.8186 | 0.0474 | 0.0070 |

Note: SRF = Subretinal Fluid; CST = Central Subfield Thickness. Pearson correlation values represent the absolute correlation.

Table 6: Predictive importance of clinical features for post-treatment VA in INDEX using different feature selection methods. Features selected for the proposed approach are indicated in bold.

| Feature | Feature Selection Metrics | | | |
|---|---|---|---|---|
| | Pearson Correlation | F-test ANOVA | Mutual Information | Random Forest Importance |
| *Primary Clinical Parameters* | | | | |
| **Baseline VA (ETDRS)** | 0.5562 | 8.0615 | 0.4362 | 0.3483 |
| **DM duration (years)** | 0.3915 | 3.2587 | 0.1958 | 0.0978 |
| | | | | |
| *Systemic Comorbidities* | | | | |
| Heart Disease (yes/no) | 0.5131 | 6.4323 | 0.1697 | 0.0669 |
| Amputation (yes/no) | 0.3410 | 2.3685 | 0.0000 | 0.0023 |
| Kidney Disease (yes/no) | 0.1037 | 0.1956 | 0.0000 | 0.0099 |
| Stroke (yes/no) | 0.0370 | 0.0247 | 0.0000 | 0.0033 |
| | | | | |
| *Treatment History* | | | | |
| Baseline Nonresponder Ozurdex (yes/no) | 0.0688 | 0.0855 | 0.0000 | 0.0028 |
| DM Treatment (tablet/insulin/combination) | 0.3122 | 1.9445 | 0.1100 | 0.0453 |
| Baseline Previous Anti-VEGF Injections (n) | 0.2739 | 1.4599 | 0.0000 | 0.1071 |
| | | | | |
| *Clinical Measurements* | | | | |
| Baseline IOP (mmHg) | 0.3471 | 2.4658 | 0.0864 | 0.0983 |
| Baseline CST (µm) | 0.0943 | 0.1616 | 0.2059 | 0.0394 |
| HbA1c (%) | 0.0196 | 0.0069 | 0.0055 | 0.0772 |
| | | | | |
| *Demographic Factors* | | | | |
| Age (years) | 0.3271 | 2.1568 | 0.0000 | 0.0549 |
| DM type | 0.1957 | 0.7170 | 0.0000 | 0.0024 |
| Gender | 0.3329 | 2.2432 | 0.0387 | 0.0179 |

Note: CST = Central Subfield Thickness; HbA1c = Glycated Hemoglobin. Pearson correlation values represent the absolute correlation.

