# OpenReview forum: "Enhancing Post-Treatment Visual Acuity Prediction with Multimodal Deep Learning on Small-scale Clinical and OCT Datasets"
_MIDL.io/2025/Conference — MIDL 2025 Poster_

### Official Review · Reviewer_cNDq · 2025-02-18

**Confidence:** 5
**Preliminary Rating:** 3
**Final Rating:** 4

**Summary:**

This paper proposes a multimodal deep learning framework that integrates optical coherence tomography (OCT) scans with patient clinical data to improve predictions of post-treatment visual acuity (VA) in diabetic macular edema (DME) patients. While the study explores a relevant clinical problem and provides some feature importance analysis, it lacks novelty and proper ablation studies to validate the approach.

**Strengths:**

- Integrating OCT scans with clinical data is a promising approach.
- The authors provide an analysis of the importance of features, which helps understand the model’s decision-making process. Additionally, the availability of code enhances reproducibility.

**Weaknesses:**

- The paper lacks novelty; the proposed multimodal approach is the most straightforward the authors could opt for.
- It is not explicitly stated which clinical features are included, and given their lower dimensionality compared to the B-scan representation, their contribution to the final prediction is uncertain.

**Detailed Comments:**

- The abstract does not clearly describe the proposed method, leaving key aspects of the approach unclear.
- The study does not include a proper ablation analysis, making it difficult to assess the individual contributions of OCT and clinical features. - For the clinical data-based baselines, using the same encoder, like the multimodal approach would make more sense than using random forest methods.
- Table 2 does not list the input data used for each model, making it difficult to follow.
- An error scatter plot could provide better insights into potential biases, as the model may primarily predict population means rather than capture meaningful variations.
The interpretability analysis through GradCAM only provides insights into the importance of the visual features. Based on the reported results, it is impossible to quantify the importance of the clinical features.

**Justification Of The Final Rating:**

I appreciate the authors' discussion and the improvements in the manuscript that greatly enhanced its clarity. The presentation of the results is easier to follow now, and the baseline comparisons are more meaningful.

**Justification Of The Preliminary Rating:**

While the paper presents a relevant and clinically meaningful problem, it lacks novelty and omits key methodological details, particularly regarding the clinical features and the absence of an ablation study on the proposed method. Addressing these concerns would significantly strengthen the work.

**Questions To Address In The Rebuttal:**

N/A

---

> ### Author Response · Authors · 2025-03-07
> **Authors Response to Reviewer cNDq**
>
> We thank the reviewer for the valuable comments and suggestions. The manuscript modifications appear in red in the new submission.
>
> **Regarding lack of novelty:** We acknowledge the concern about simplicity. Given extremely small datasets, our design balances performance with overfitting risk. More complex architectures would likely struggle to generalize with limited data.
>
> Our novelty lies in demonstrating that a carefully designed multimodal approach can predict post-treatment VA with extremely limited data (n=35 for DIME, n=20 for INDEX). Our cross-modal attention mechanism enables integration of heterogeneous data types and improves performance over single-modality approaches. Limited literature exists on post-treatment VA prediction using multimodal deep learning with small datasets.
>
> **Regarding clinical feature usage:** We've updated the manuscript to explicitly state selected features: baseline VA (ETDRS), age (years), and DM duration (years) for DIME; baseline VA (ETDRS) and DM duration (years) for INDEX. We've clarified their integration through our cross-modal attention mechanism (Equation 1), which modulates image features using clinical parameters.
>
> **Regarding abstract clarity:** We've revised the abstract to provide more detailed description of our multimodal framework, explicitly describing how the model processes and combines different data modalities while maintaining conciseness.
>
> **Regarding ablation analysis:** We've conducted a study evaluating contributions of OCT imaging and clinical features, including models trained using only OCT features, only clinical features, and our multimodal approach. Results in Table 3 demonstrate that while clinical features provide strong predictive power, adding OCT imaging improves performance.
>
> We've replaced gradient-boosting with a neural network using the same architecture as our clinical network in our multimodal approach for fairer comparison. Updated results highlight relative performance of different modalities under a consistent framework.
>
> **Regarding table issues:** We've updated Table 2 to explicitly indicate input data for each model, adding a column specifying whether models utilize OCT images, clinical features, or both. We've improved formatting of all tables for clarity.
>
> **Regarding biases and feature importance:** Figure 3(a) and 3(b) visualize error distributions across different VA ranges, demonstrating systematic biases. Our analysis quantifies these biases, showing regression toward the mean in DIME and under-prediction bias in higher VA ranges for INDEX.
>
> Our feature importance analysis through Integrated Gradients quantified relative contributions of imaging and clinical features. OCT features provided the strongest predictive signal (mean=8.07±4.21), complemented by baseline VA (mean=5.78±1.95). Our ablation study further quantifies clinical feature impact by measuring performance degradation when baseline VA is replaced.

---

### Official Review · Reviewer_XW7x · 2025-02-19

**Confidence:** 5
**Preliminary Rating:** 4
**Recommendation:** Oral

**Summary:**

The paper presents a multimodal deep learning approach to predict post-treatment visual acuity (VA) in Diabetic Macular Edema (DME) patients by integrating Optical Coherence Tomography (OCT) images with clinical data. The model is designed to address challenges posed by limited clinical datasets and aims to enhance predictive accuracy for post-treatment visual outcomes.
Dataset:
Two small-scale datasets from clinical trials:
DIME study (n=35) – Treatment-naïve DME patients receiving anti-VEGF therapy.
INDEX study (n=20) – Chronic DME patients receiving sustained-release implants.
Data Preprocessing:
OCT standardization: Resizing and normalization to a consistent resolution.
Feature selection for clinical data using statistical and machine learning approaches (Pearson correlation, Random Forest importance, ANOVA, and Mutual Information analysis).
Model Architecture:
EfficientNet-B0-based CNN for OCT feature extraction.
Feedforward network for clinical data processing.
Cross-modal attention module to enhance fusion between imaging and clinical features.
Final VA prediction using a fully connected layer with LayerNorm and GELU activation.
Evaluation Metrics:
Mean Absolute Error (MAE), Root Mean Square Error (RMSE), and R².
Stratified 5-fold cross-validation based on post-treatment VA categories.

**Strengths:**

The paper presents a well-structured and scientifically rigorous approach to post-treatment visual acuity (VA) prediction in diabetic macular edema (DME) patients using a multimodal deep learning framework. One of its strongest contributions is the use of cross-attention mechanisms to modulate OCT image features based on clinical data, rather than using simple feature concatenation. This approach allows the model to prioritize clinically relevant biomarkers and enhances interpretability, making it more suitable for real-world clinical applications.

Another key strength is the paper's focus on small-scale clinical datasets, addressing a major limitation in medical AI research where large, well-curated datasets are often unavailable. The authors implement statistical feature selection methods (Pearson correlation, Random Forest importance, ANOVA, and Mutual Information analysis) to ensure that the model only incorporates clinically meaningful variables, making the model more robust despite limited training data. The use of stratified 5-fold cross-validation helps mitigate overfitting concerns and ensures balanced evaluation across different VA levels.

Additionally, the paper includes thorough baseline comparisons, testing against both clinical-only models (linear regression, random forest, gradient boosting) and imaging-only models (ResNet-50, EfficientNet-B0). The proposed multimodal approach outperforms all baselines, demonstrating its effectiveness. The inclusion of explainability techniques (Grad-CAM for OCT features and Integrated Gradients for clinical data) further improves transparency and clinical interpretability.

**Weaknesses:**

🔹 The paper evaluates the model using stratified 5-fold cross-validation, which ensures balanced validation across different VA levels. However, there is no fully independent test dataset used to assess true generalizability. Cross-validation alone cannot fully substitute for external validation, as models can still overfit to the dataset's specific distribution. Testing on an external dataset from another clinical center would strengthen the model’s clinical applicability.

🔹 The authors manually removed clinical features that were not ranked high by statistical tests.
This means the deep learning model was never given the chance to learn relationships from all available clinical data.
If a feature had a complex non-linear interaction with visual acuity, but was not statistically significant in a linear test, it might have been wrongly excluded.

🔹 The study is based on two small-scale datasets (DIME: 35 cases, INDEX: 20 cases), which limits statistical power. While the authors implement feature selection and transfer learning to mitigate overfitting, the model’s generalization remains uncertain due to the high variance in the INDEX dataset (R² = 0.61 ± 0.36). Small datasets also increase the risk of model performance being influenced by noise rather than true predictive patterns.

**Detailed Comments:**

🔹 Concern: The authors manually pre-selected clinical features using statistical methods (Pearson correlation, ANOVA, Random Forest importance, and Mutual Information) instead of allowing the deep learning model to automatically learn feature importance. This could introduce bias by removing features that might have been useful in a deep learning framework, especially for capturing non-linear relationships.
🔹 Suggestion: Train an alternative model with all clinical features included and use deep learning feature attribution methods (SHAP, Integrated Gradients) to compare whether deep learning selects the same features as statistical methods or not.

🔹 Concern: The model is only evaluated using stratified 5-fold cross-validation, and there is no independent test set to verify generalization. Without testing on completely unseen data, the reported results may be influenced by dataset-specific biases.
🔹 Suggestion: The authors should validate the model on an independent dataset (e.g., data from another clinical center) to ensure robustness. If an external test set is unavailable, they could hold out a portion of the dataset as a dedicated test set rather than relying only on cross-validation.

**Justification Of The Preliminary Rating:**

The paper presents a well-structured multimodal deep learning approach for predicting post-treatment visual acuity (VA) in diabetic macular edema (DME) patients, integrating OCT imaging and clinical data. The model incorporates cross-modal attention mechanisms, making it more sophisticated than simple feature concatenation. The results demonstrate clinically meaningful predictive performance, with stratified 5-fold cross-validation showing strong correlations between model predictions and actual VA outcomes. The study is particularly valuable because it tackles small-scale clinical dataset challenges by applying statistical feature selection and transfer learning, ensuring robust model training despite limited samples.

However, significant concerns remain regarding feature selection and generalization. The paper relies entirely on statistical feature selection (Pearson correlation, ANOVA, Random Forest importance, Mutual Information), but does not explore whether deep learning could have identified better predictive features on its own. While Integrated Gradients is used for post-hoc feature importance interpretation, the study does not compare statistical rankings with deep learning-derived importance scores, leaving open the possibility that important clinical features were excluded due to pre-selection biases. Additionally, the sensitivity of the model to different feature selection strategies is not tested, meaning its robustness across different datasets remains uncertain.

**Questions To Address In The Rebuttal:**

Clarity on how Does Pre-Selecting Clinical Features Impact Model Performance Compared to Letting Deep Learning Select Features Automatically? how to make sure there is not feature bias
Have the authors tested different feature selection strategies (e.g., recursive feature elimination, L1 regularization) to verify feature robustness? if yes, what was the results, if not why they did not?

**Special Issue:**

No

---

> ### Author Response · Authors · 2025-03-07
> **Authors Response to Reviewer XW7x**
>
> We thank the reviewer for the valuable comments and suggestions. The manuscript modifications appear in red in the new submission.
>
> **Regarding lack of independent test dataset:** We acknowledge the reviewer's concern and agree that external validation would strengthen the model's generalizability. However, due to the extremely small sample sizes of our datasets, creating a fully independent test set would significantly reduce the training data and impact model performance. Instead, we employed stratified 5-fold cross-validation to ensure that each fold contained a representative distribution of VA outcomes, which a small hold-out set could not guarantee.
>
> It is common in the literature to report cross-validation performance when working with small datasets, as it allows for a more reliable estimate of model performance than a single small test set. Nevertheless, we have expanded the limitations section to explicitly acknowledge this constraint and emphasize the need for future studies with larger datasets to incorporate external validation. This study serves as a proof-of-concept, demonstrating the feasibility of multimodal approaches even in data-limited clinical settings.
>
> **Regarding manual feature selection and potential bias:** We appreciate the reviewer's concern regarding feature selection and the potential exclusion of clinically relevant non-linear interactions. To address this, we conducted an ablation study comparing our selective approach with a model that incorporates all available clinical features. The results showed significant performance degradation when using all features (MAE increased by 1.78 and 2.36 ETDRS letters for DIME and INDEX, respectively), confirming that our feature selection approach effectively mitigates overfitting on small datasets.
>
> To minimize feature selection bias, we employed multiple complementary techniques (Pearson correlation, random forest importance, ANOVA, and mutual information analysis) to capture both linear and non-linear relationships from different statistical perspectives. We did not implement recursive feature elimination or L1 regularization specifically because our extremely small sample sizes make such iterative techniques less stable and potentially more prone to overfitting to noise rather than signal. The consistent performance degradation observed when using all features validates our approach, suggesting that in small dataset scenarios like ours, carefully selected features based on multiple statistical methods outperform models with access to all features.
>
> **Regarding small dataset size:** We acknowledge the limitations associated with our small dataset sizes and the impact on statistical power and model variance, particularly in the INDEX dataset. To mitigate these challenges, we employed several strategies: (1) transfer learning with an ImageNet-pretrained EfficientNet-B0 backbone, fine-tuning only the last two blocks to prevent overfitting, (2) regularization through dropout (rate = 0.3) and early stopping, and (3) dataset-specific data augmentation techniques. These approaches are widely used in medical imaging research to improve generalization in data-limited scenarios.

---

> > ### Comment · Reviewer_XW7x · 2025-03-14
> >
> > Thank you for addressing my questions and considering my concerns in your rebuttal. I have no further inquiries and will maintain my initial rating

---

### Official Review · Reviewer_P4Nu · 2025-02-20

**Confidence:** 4
**Preliminary Rating:** 3
**Recommendation:** Poster
**Final Rating:** 4

**Summary:**

In this paper, they present a new method for predicting post-treatment visual acuity (VA) -- the prognosis of VA -- from OCT B-scans and patient clinical data. Their methodological contributions are (1) a hybrid neural network to handle the multimodal inputs, (2) application to small datasets, and (3) selection of clinical features. They can show better performance compared to single modality methods.

**Strengths:**

- Simple yet effective multimodal architecture for predicting VA from OCT B-scans and clinical features.
- Works for small datasets: only blocks 6 and 7 of an EfficientNet-B0 (pre-trained on ImageNet) were fine-tuned.
- The evaluations highlight various aspects: regression performance, Grad-CAM, error analysis across different post-treatment VA ranges, mean feature attribution with Integrated Gradients.
- Publicly available source code to reproduce the results (data is available upon request).

**Weaknesses:**

- Why are the comparison methods in Table 2 only single-modality methods? The authors write in the introduction that the gap they address is the prediction of post-treatment VA with limited datasets. They point out that related methods from the literature "often rely on large, well curated datasets, limiting their generalizability to real-world clinical settings." My question is why the authors did not include the related works of Wen et al. and Wang et al. in Table 2?
- Feature selection: Why did the authors select only a few clinical features out of several (Table 3: 3/11 for DIME, Table 4: 2/15 for INDEX)? I understand that they measured several feature selection metrics by the predictive performance of these features *alone* for post-treatment VA, and selected only those with the best values. But what about the predictive performance of the combination of clinical features and OCT B-scans? So why didn't the authors do the experiment where they gave the model *all* the available features and let it *automatically* select their importance? Then they could also compare this performance with their feature selection approach.
- Grad-CAM also appears to indicate "regions of intraretinal fluid accumulation" for DIME, where the corresponding clinical feature would be intraretinal fluid (IRF).
- I also wonder how much the performance would deteriorate if the authors excluded "Baseline VA (ETDRS)". So how well can a model predict "Baseline VA (ETDRS)" (current VA state) from the OCT B-scan and the rest of the clinical features? There is already an indication in Figure 4 that this feature is the most important of the 2/3 selected clinical features. However, if a model cannot predict the current VA state, how well can it predict post-treatment VA?

**Detailed Comments:**

- I think Figure 1 needs some refinement. The font size is too small to read it without zooming in. Also, in the neural network visualizations, the circles are sometimes squeezed.
- Since each fold in Figure 4 seems to focus on very different features, I'm interested in the performance metrics for each fold individually (you only report means in Table 2).

**Justification Of The Final Rating:**

I thank the authors for the discussion. They clarified all my open questions and especially addressed my main concerns regarding comparisons with other multi-modal approaches and their feature selection. Therefore, I am happy to raise my final rating to "weak accept".

**Justification Of The Preliminary Rating:**

This work is an important step toward effective multimodal small-scale post-treatment prediction. However, I have some concerns about one aspect of the paper. If I understand correctly, this paper is about two aspects: (1) multimodal deep learning and (2) small scale datasets. In my opinion, the authors address point (1) well in the paper by comparing the performance of their model with single modality methods. However, regarding point (2), I'm missing experiments comparing it to other multimodal models trained only on their small datasets.

**Questions To Address In The Rebuttal:**

Please address the points in the Weaknesses and Detailed Comments section.

**Special Issue:**

No

---

> ### Author Response · Authors · 2025-03-07
> **Authors Response to Reviewer P4Nu**
>
> We thank the reviewer for the valuable comments and suggestions. The manuscript modifications appear in red in the new submission.
>
> **Regarding comparison methods in Table 2:** We have enhanced Table 2 by incorporating comparisons against the works of Liu et al. and Wen et al., and improved clarity by specifying data types used for each model. These multimodal comparisons are based on their reported performance metrics on similar post-treatment VA prediction tasks from their respective publications, as implementing their models on our specific datasets wasn't feasible within the rebuttal timeframe and due to limited code availability. While not a direct comparison on identical datasets, this addition provides important context on how our approach compares to other multimodal methods addressing similar clinical problems.
>
> **Regarding our feature selection approach:** We limited selection to one feature per 10 observations following established guidelines for small datasets (Peduzzi et al., 1996) to prevent overfitting, reduce model complexity, and improve generalizability given our limited sample size. As suggested, we have included an ablation study using all available clinical features to demonstrate the tradeoffs between comprehensive feature inclusion and our more selective approach optimized for small datasets.
>
> **Regarding the Grad-CAM observation:** We agree with the reviewer's observation that Grad-CAM highlights regions of intraretinal fluid accumulation, which aligns with the intraretinal fluid (IRF) clinical feature. This provides additional validation that our model is attending to anatomically and clinically relevant features in the OCT images, reinforcing the effectiveness of our multimodal approach.
>
> **Regarding the importance of Baseline VA:** We agree that Baseline VA is a strong predictor of post-treatment VA, which aligns with clinical understanding---since baseline functional status is a key determinant of treatment response and is usually measured in practice. To assess our model's dependence on Baseline VA, we have conducted an ablation study where we removed it and instead used the next highest-ranked clinical predictor (IRF Cysts for DIME and Baseline IOP for INDEX). This resulted in reduced performance across both datasets, confirming the critical role of Baseline VA in predicting post-treatment VA. While predicting Baseline VA from OCT and other clinical features is an interesting idea, our study focuses specifically on post-treatment VA prediction. The fact that Baseline VA is not fully explained by imaging and other clinical variables reinforces its independent importance in the model.
>
> **Regarding Figure 1 refinement:** We have increased all font sizes while maintaining the figure's layout within space constraints and corrected the distortion in the neural network visualizations to ensure that the component circles maintain their intended proportions, significantly improving clarity and legibility.
>
> **Regarding per-fold performance:** In response to interest in the per-fold performance metrics, we have included individual per-fold performance metrics in the appendix Table 4, providing a clearer view of the variability across folds to complement the aggregated results in Table 2.

---

> > ### Comment · Reviewer_P4Nu · 2025-03-11
> > **Open questions**
> >
> > Thank you for addressing my concerns and for the additional effort you put into your submission. However, I still have some open questions regarding the revision:
> >
> > **Table 2, metric values**: I checked the basline papers and saw that they both report logMAR values. I'm not familiar with the Mean Refractive Spherical Equivalent (logMAR) and how to convert it to the Mean Absolute Error (MAE) and Root Mean Squared Error (RMSE) in ETDRS letters that the authors reported in Table 2. Can the authors please explain how they determined these values from the baseline papers?
> >
> > **Table 2, multi-modal baselines**: I saw that the authors replaced Wang et al. with Liu et al. without highlighting it in the paper. Wasn't it possible to compare with Wang et al.? And why didn't the authors highlight and explain the exchange?

---

> > ### Author Response · Authors · 2025-03-11
> > **Authors Response To Reviewer P4Nu open questions**
> >
> > We sincerely thank the reviewer for their detailed examination of our revised manuscript and for highlighting these important technical questions about our methodology and reporting.
> >
> > **Regarding the logMAR to ETDRS conversion**
> > We used the standard equation for converting logMAR values to ETDRS letter scores:
> > $$
> > \text{ETDRS Letters} = 85 - (50 \times \text{logMAR})
> > $$
> >
> > This equation provides the absolute ETDRS letter score for a given logMAR value. For our error metrics (MAE and RMSE), we first calculated these metrics using the original logMAR values from the baseline papers, and then converted these error values to ETDRS letters to ensure consistent comparison.
> >
> > Since each 0.02 logMAR step corresponds to 1 ETDRS letter, a 1.0 logMAR change equates to 50 ETDRS letters (i.e., $1.0 / 0.02 = 50$). Therefore, to express error values in ETDRS letters, we multiplied the logMAR errors by 50.
> >
> > For example:
> > - An MAE of 0.13 logMAR translates to 6.5 ETDRS letters ($0.13 \times 50 = 6.5$).
> > - An RMSE of 0.20 logMAR corresponds to 10.0 ETDRS letters ($0.20 \times 50 = 10.0$).
> >
> > **Reference:**
> > Beck, R.W., Moke, P.S., Turpin, A.H., Ferris, F.L., SanGiovanni, J.P., Johnson, C.A., Birch, E.E., Chandler, D.L., Cox, T.A., Blair, R.C., & Kraker, R.T. (2003). *A computerized method of visual acuity testing.* American Journal of Ophthalmology, 135(2), 194-205.
> >
> > ---
> >
> > **Regarding replacing Wang et al. with Liu et al.**
> >
> > We acknowledge that we did not explicitly highlight this change in our initial revision, and we appreciate the reviewer bringing this to our attention. Upon careful analysis of both papers, we opted to reference Liu et al. instead of Wang et al. for several methodological reasons. While both papers utilize OCT imaging, Liu et al.'s framework provides a more appropriate comparison for our work.
> >
> > Specifically, Liu et al.'s ensemble machine learning system combines multiple deep learning models with various classical ML models in a comprehensive fusion architecture, which more closely aligns with our hybrid neural network architecture. In contrast, Wang et al. employed a two-stage pipeline where a semi-supervised network was first used for retinal segmentation, followed by a separate XGBoost algorithm for VA prediction.
> >
> > Furthermore, Liu et al. clearly reported their visual acuity prediction results using MAE and RMSE values in logMAR units (0.13 and 0.20, equivalent to 6.5 and 10.0 ETDRS letters), which allowed for direct comparison with our results. While Wang et al. reported MAE and RMSE values (0.106 and 0.141), they did not explicitly specify the units used. If these were in logMAR units, they would be equivalent to 5.3 and 7.05 ETDRS letters (worse than our approach). However, without explicit unit specification in Wang et al., direct comparison with our results becomes challenging.
> >
> > Given these considerations, Liu et al. provided a more transparent methodological framework with clearly specified metrics that facilitated meaningful comparison, making it a more appropriate reference for benchmarking against our proposed approach.

---

### Author Rebuttal · Authors · 2025-03-07

**Rebuttal:**

We thank all reviewers for their insightful comments which have helped strengthen our manuscript. The modifications appear in red in the revised submission.

We have addressed the major points raised across reviews:

**Enhanced comparison with state-of-the-art methods** by including relevant multimodal approaches in Table 2.

**Improved clarity of our feature selection methodology** as well as greater clarity of clinical features used, as well as justified our approach with ablation studies.

**Added detailed ablation studies** demonstrating the benefit of our feature selection approach, replacing baseline VA with a different feature and performance decrease without the cross-modal attention mechanism.

**Clarified model performance and variability** with per-fold metrics in the appendix.

**Improved visual clarity** with larger fonts in figures and better table formatting.

Our work demonstrates that carefully designed multimodal architectures can achieve clinically meaningful prediction accuracy even with limited datasets typical in real-world settings, offering practical value for personalized DME management.

**Supporting Material:**

/attachment/671294b96ce9e7b43d130ea9c31b8b2e261d380e.pdf

---

### Meta-Review · Area_Chair_sXsh · 2025-03-23

**Recommendation:** Accept (Poster)
**Confidence:** 4

**Metareview:**

The paper presents a multimodal deep learning approach integrating OCT B-scans and clinical data to predict post-treatment visual acuity (VA) in diabetic macular edema (DME) patients, focusing on small dataset constraints. Strengths include an effective hybrid neural network, cross-modal attention for feature fusion, and strong performance over single-modality baselines. However, reviewers highlight key concerns: the lack of external test data for generalization, the exclusion of potentially relevant clinical features through manual selection instead of learning-based approaches, and limited comparisons with other multimodal models.Based on the rebuttal  the paper may be considered for acceptance.